# Pork Liver Decomposition Product May Improve Frontal Lobe Function in Humans—Open Trial

**DOI:** 10.3390/brainsci14060586

**Published:** 2024-06-07

**Authors:** Miiru Suzuki, Ikuya Sato, Masatsugu Sato, Hideki Iwasaki, Takahiro Saito, Masahiko Kimura, Kenichi Sako, Tomoji Maeda, Hisao Haniu, Tamotsu Tsukahara, Yoshikazu Matsuda

**Affiliations:** 1Division of Clinical Pharmacology, Graduate School of Pharmaceutical Science, Nihon Pharmaceutical University, Ina 362-0806, Japan; eternal_peace0222@hotmail.com (M.S.); msato@gpro.com (M.S.); 217008@ym.nichiyaku.ac.jp (H.I.); ganwxyzw@gmail.com (T.S.); m-kimura@jichi.ac.jp (M.K.); sakok@nichiyaku.ac.jp (K.S.); t-maeda@nichiyaku.ac.jp (T.M.); 2Institute for Biomedical Sciences, Interdisciplinary Cluster for Cutting Edge Research, Shinshu University, Matsumoto 390-8621, Japan; hhaniu@shinshu-u.ac.jp; 3Department of Pharmacology and Therapeutic Innovation, Graduate School of Biomedical Sciences, Nagasaki University, Nagasaki 852-8521, Japan; ttamotsu@nagasaki-u.ac.jp

**Keywords:** Porcine Liver Decomposition Product, Revised Hasegawa Dementia Scale, verbal fluency task, frontal lobe function

## Abstract

Porcine Liver Decomposition Product (PLDP) was obtained by treating pig liver homogenate with protease and filling it into capsules. We have already confirmed from three clinical trials that PLDP enhances visual memory and delays memory recall, and we believe that its activity is due to various phospholipids, including phosphatidylcholine (PC). In this study, we clinically evaluated PLDP for depressive symptoms caused by a decline in cognitive function. This clinical trial was conducted using the Revised Hasegawa Dementia Scale (HDS-R). The HDS-R (maximum score is 30 points) is a test similar to the Mini-Mental State Examination (MMSE), which is commonly used in Japan. Dementia is suspected if the score falls below 20 on the HDS-R. Additionally, in a previous clinical trial, there was no change in scores in the placebo group after three doses of the HDS-R. In order to clearly confirm the effectiveness of PLDP, this study was conducted under stricter conditions (HDS-R points of 15 to 23) than previous clinical trials (all participants had scores of 20 or higher). Therefore, from ethical considerations, a clinical trial was conducted using the scores before PLDP administration as a control. In this study, PLDP was administered orally at 4 capsules per day, and the HDS-R was confirmed 2 and 4 weeks after administration. A significant increase in HDS-R scores was observed at 2 and 4 weeks after PLDP administration. Additionally, regarding each item of the HDS-R, PLDP significantly increased 2 and 4 weeks after oral administration for the question items assessing delayed recall, and the question item assessing verbal fluency tasks was recognized. From the above results, we confirmed the reproducibility of the effect of PLDP in improving the delayed recall of verbal memories. Furthermore, increasing scores on verbal fluency tasks suggest that PLDP may enhance frontal lobe function and prevent or improve depressive symptoms. The effects observed in this study may differ from the mechanisms of action of existing antidepressants, and we believe that this may lead to the discovery of new antidepressants.

## 1. Introduction

Porcine Liver Decomposition Product (PLDP) is a functional food prepared by treating porcine liver homogenates with protease and filling the capsules after high-pressure steam sterilization. It is rich in various phospholipids, including phosphatidylcholine (PC) (Table 1). Matsuda et al. conducted an open trial and double-blind, placebo-controlled study and found that oral administration of PLDP for 2 and 4 weeks was effective in participants with a score of 20 or higher on the Hasegawa Dementia Scale-Revised (HDS-R), which was proven to significantly increase scores [1]. Matsuda et al. also established that the oral administration of PLDP significantly increased visual memory and delayed recall scores on the Wechsler Memory Scale-Revised Edition (WMS-R) in healthy adults over 40 years of age compared to placebo [2]. Based on these reports, PLDP is expected to have the potential to improve age-related decline in memory acquisition and regeneration. The mechanism underlying the promotion of memory acquisition and delayed recall by PLDP is the biosynthesis of PC into acetylcholine (Ach), which promotes memory acquisition [3]. However, its exact mechanism of action remains unclear.

Contrarily, Tsukahara et al. reported that the lipid fraction contained in PLDP might increase anti-inflammatory-type microglia and suppress microinflammation in the brain [4,5]. Recent research has indicated that microinflammatory responses in the brain are involved in the pathology of many psychiatric disorders such as depression [6]. This suggests that PLDP may affect psychiatric disorders, in addition to promoting memory and delayed recall. Furthermore, we hope that the clinical and basic pharmacological evaluation of PLDP will lead to the elucidation of the active substances contained in PLDP and the exact pathogenesis of psychiatric disorders.

The HDS-R is a cognitive function test that is similar to the Mini-Mental State Examination (MMSE), which is commonly used in Japan because it is composed of items related to orientation and various qualitatively different memories, due to its short measurement time and simplicity. The HDS-R questions are presented in Table 2. The HDS-R is a 30-point test consisting of nine items, including age, orientation, verbal memory, attention, delayed recall, visual memory, and verbal fluency. The meaning of the HDS-R is as follows: An HDS-R score of 20 or less indicates a high suspicion of dementia. The average score for people without dementia is approximately 24 points, approximately 19 points for people with mild dementia, approximately 15 points for people with moderate dementia, and approximately 10 points for people with severe dementia [7,8]. We believe that confirming the effects of PLDP on each question item is meaningful for elucidating its clinical effects and mechanisms of action.

In this study, we examined the effects of each question item on the HDS-R using doses of PLDP that were effective in previous clinical trials. Additionally, the HDS-R results from a double-blind, placebo-controlled study of PLDP indicated that the participants’ initial HDS-R values were high, and no significant difference in PLDP was observed compared to the placebo [1]. Therefore, we decided to conduct this study using participants whose HDS-R scores ranged from 15 to 23 points during the pre-screening period. We conducted an open test to examine the HDS-R total score and question item scores at 2 and 4 weeks after the oral administration of PLDP. Furthermore, since it is thought that the participant group in this study may have mild cognitive impairment, the doctors who conducted the clinical trial made ethical judgments and conducted the trial safely and ethically.

## 2. Materials and Methods

### 2.1. Ethical Approval

This study was conducted according to the Human Participants Ethics Guidelines of the Japanese Ministry of Health, Labor, and Welfare. All participants provided written informed consent according to the Declaration of Helsinki. The research protocol was approved by the Institutional Ethics Committee of the Nippon Pharmaceutical University (approval date: 11 October 2015, NPUEC 20151011) and the Abe Clinic Institutional Review Board (approval date: 12 August 2015, AbeEC 20150812). This study was registered with the University Hospital Medical Information Network (UMIN000021530) and conducted according to the CONSORT guidelines. Furthermore, to ensure the objectivity of this research, we outsourced the testing to a contract research organization (Relife Co., Ltd., Tokyo, Japan).

### 2.2. PLDP

PLDP is a product of protease treatment of Sugar Pork liver (Sugar Lady Cosmetics Co., Ltd., Tokyo, Japan). It can be added to capsules after high-pressure steam sterilization. PLDP was synthesized by YAEGAKI Bio-Industry, Inc. (Himeji, Japan). PLDP primarily includes phospholipids such as PC. The components and amounts of PLDP administered daily are listed in Table 1. Phospholipid levels were measured after sterilization with PLDP. Phosphatidylcholine, which we consider a marker, had an 18% change over 4 months (at 40 °C, 75% RH) in a stability study.

### 2.3. Study Design

The results of the PLDP clinical trial are presented in Table 3. All studies shown in Table 3 had a 4-week study period. The HDS-R and WMS-R were performed before administration and at 2 and 4 weeks after administration. In addition, studies 1 and 2 were conducted on participants with HDS-R scores of 20 or higher. As depicted in Table 3, 4 capsules per day of PLDP significantly increased the HDS-R and WMS-R scores. Moreover, no significant changes were observed in the placebo group in two double-blind placebo-controlled trials. Therefore, due to ethical considerations, this was an open study with no control group. The data were evaluated based on the change in the values before PLDP administration.

Based on previous reports [1], the HDS-R was conducted before, 2 weeks after, and 4 weeks after oral administration of PLDP.

### 2.4. Recruitment of Participants

This study was conducted at Mori Internal Medicine Clinic (Tokyo, Japan). The number of participants was determined by referring to the number of participants in the study, as depicted in Table 2. Informed consent was obtained from 13 participants, and the study was conducted on 11 participants based on the inclusion and exclusion criteria described below:

[Inclusion criteria]

Those who are 20 years of age or older on the date of obtaining consent for the test.Those with a score of 15 or more and 23 or less on the HDS-R.A person who can explain the purpose and content of the test before the test and obtain written consent from the participant.

[Exclusion criteria]

Those who must continue to take anti-dementia drugs or those who will start taking anti-dementia drugs (this study will not be successful).Those taking medicines that are expected to affect the HDS-R (this study will fail).Individuals with a history of brain disease or complications that affect the HDS-R (this study will not be successful).People who have difficulty swallowing (as this product may cause aspiration).Not understanding the purpose of this study (because it may lead to adverse effects).Participants whose treatment may change during this study (because the study will be unsuccessful).Difficulty conducting intelligence tests (because the test will fail).Allergic to pork liver (to ensure the safety of the participants).Persons with malignant tumors or complications, such as severe heart, kidney, or liver disorders; respiratory diseases; or circulatory diseases (to ensure the safety of the participants).Other persons judged inappropriate for inclusion by the doctor in charge of the study.

### 2.5. Statistical Analysis

The obtained results were expressed as mean value ± standard deviation, and statistically significant differences were analyzed by unpaired *t*-test and Welch’s *t*-test after analysis of variance.

## 3. Results

Figure 1 presents an overview of this study. The contents of the study were explained to 13 potential test participants, and their consent to participate in the study was obtained. Subsequently, one participant was excluded from the study because he was taking medication that affected cognitive function, and another participant withdrew consent. Consequently, we decided to conduct the study on 11 participants aged 65–86 years who had HDS-R scores of 15–23 and provided written informed consent.

Table 4 presents the participants’ age, HDS-R score, and the name of disease for which they visited the clinic. The participants who participated in the study were five males and six females. A doctor confirmed that the patient’s disease did not affect their cognitive function. The mean age of the participants was 77 ± 3 years. The HDS-R score of the 11 participants was 21.4 ± 0.7.

Figure 2 presents the actual measured values and changes in the HDS-R before, 2 weeks after, and 4 weeks after the oral administration of PLDP in participants with HDS-R scores of 15–23 points. Four weeks after oral administration, the scores were 21.4 ± 2.3 before feeding, 24.0 ± 4.4 at 2 weeks, and 25.3 ± 3.6 4 weeks after oral administration. Additionally, a significant increase in the change was observed with oral administration of PLDP (2.6 ± 2.7 after 2 weeks and 3.9 ± 3.0 after 4 weeks).

Figure 3 presents the changes in each question (Q1-9) in the HDS-R at 2 and 4 weeks after the oral administration of PLDP. Two weeks after the oral administration of PLDP, a significant increase was observed in the Q7 score of 1.1 ± 1.4 and the Q9 score of 1.1 ± 1.6. Furthermore, 4 weeks after the oral administration of PLDP, the Q7 score was 1.6 ± 2.2 and the Q9 score was 1.2 ± 1.7, which had significantly increased. Regarding other question items, there was an upward trend in Q2 and Q6 scores 2 weeks after oral administration of PLDP and an upward trend in Q2 scores 4 weeks after oral administration; however, there were no statistically significant differences.

## 4. Discussion

In this study, the oral administration of PLDP significantly increased HDS-R scores at 2 and 4 weeks in participants with HDS-R scores of 15–23 points compared with those before administration. Furthermore, when evaluating each HDS-R question item, we established significantly increased scores for delayed recall of verbal memory and verbal fluency tasks. Matsuda et al. reported that scores increased in participants with HDS-R scores of 20 or higher after continuous oral administration of PLDP for 2 and 4 weeks [1]. In this study, we were able to confirm the effectiveness of PLDP even in participants with HDS-R scores of 15 to 23 points who may have mild cognitive impairment. Matsuda et al. also reported that continuous oral administration of PLDP for 4 weeks increased delayed memory recall and visual memory scores on the WMS-R [2]. In this study, PLDP also increased delayed verbal recall scores, similar to the results observed with the WMS-R. However, although an upward trend was observed regarding date, time orientation, and number of recitations, no significant changes were observed. Furthermore, no change was observed in the product name of Q8, which was different from the results obtained from the WMS-R. We believe that it is necessary to examine the homology of the results due to differences in the evaluation and analysis methods in the future.

The verbal fluency task (VFT), also called the word recall task, is a neuropsychological test often used in language evaluation and cognitive function screening. It is also included in the standardized test battery of the HDS-R, and Q9 corresponds to a category-fluency task (CFT). In addition to frontal function, the CFT requires the ability to recall words using meaning cues [9]. Furthermore, Audenaert et al. measured the cerebral blood flow during a CFT task using SPECT and reported that the left inferior frontal cortex, left prefrontal cortex, and right inferior frontal cortex were activated [10], whereas Gorovitch et al. confirmed the activity of the prefrontal cortex during a CFT task [11]. In this study, PLDP intake significantly increased the Q9 scores, suggesting that PLDP has the potential to increase blood flow to the frontal region and improve its function.

Research has shown that information transmission circuits centered in the amygdala and reward system circuits centered in the nucleus accumbens are intricately intertwined with neural circuits centered in the prefrontal cortex, leading to pathological conditions [12,13,14]. Furthermore, when we compared brain activity during the VFT in 10 depressed patients and 10 healthy participants matched by age and sex using functional magnetic resonance imaging, we established that depressed patients exhibited increased activation during the VFT. Functional decline in the left prefrontal cortex has also been reported [14]. Considering the above points, we believe that PLDP activates the frontal lobe function and is expected to have an antidepressant effect. It has also been suggested that this effect may be exerted through a mechanism different from that of current drugs, which act directly on neurotransmitters and their receptors. Therefore, further detailed investigations are necessary.

Regarding research on the active substance of PLDP, Tsukahara et al. reported that the lipid fraction and lysophospholipid group in PLDP suppressed inflammatory cytokines and activated oxygen release when SIM-A9 microglia were stimulated with LPS [4,5,15]. In the brain, microglia are involved in neuroinflammation and have been suggested to play an important role in forming neural circuits and maintaining neurotransmission homeostasis [16,17]. Furthermore, positron emission tomography imaging targeting activated microglia has been reported to show a positive correlation between depressive symptoms in patients with depression and microglial activation in the prefrontal cortex, anterior cingulate cortex, and hippocampus [18]. There is also a correlation between the strength of suicidal ideation and microglial activation [19]. Additionally, in a repeated social defeat stress model in rats, it was reported that bone marrow hematopoiesis is promoted via the sympathetic nervous system, immature monocytes migrate into the brain, and microglia are activated. It has been suggested that peripherally derived monocytes and intracerebral microglia mutually amplify neuroinflammation due to emotional reactions (anxiety-like behaviors) caused by social stress [20]. Therefore, it is possible that PLDP modulates the function of microglia in the brain, enhances the function of the frontal lobe, and consequently increases the CFT scores. This suggests that PLDP may have antidepressant effects, and we believe that further investigation is necessary in the future.

PLDP contains phosphatidylcholine, which is a source of choline. Mirja Kaizer Ahmmed et al. and Jing Wen et al. have reported that LPC containing ω-3 polyunsaturated fatty acids may play an important role in brain development and neuronal growth [21,22]. Murota et al. reported that PC is important for the absorption of choline and polyunsaturated fatty acids [23]. The effects observed in this study were due to the choline and polyunsaturated fatty acids contained in the PLDP. Furthermore, using the eight-way radial maze, Inoue et al. indicated that long-term administration of choline chloride promotes the acquisition and retention of learning and memory in rats, and one mechanism for this is the production of brain-derived neurotrophic factor (BDNF) in the hippocampus [24]. BDNF, which is produced by neurons and microglia in the hippocampus and cerebral cortex, plays an important role in neuronal development, such as in the differentiation, maturation, and survival of neurons; axonal elongation and synapse formation; modification of neurotransmission; neuronal regeneration; and learning and memory [25]. Inflammatory cytokines suppress the action of BDNF [26]. These reports suggest that the phospholipids contained in PLDP exert neuroprotective effects through choline and fatty acids, suppress inflammatory cytokines produced by microglia, and enhance BDNF production. Therefore, the results of this study were accurate. We believe that a detailed examination of these issues is required in the future.

These results revealed that PLDP was effective in improving the delayed retrieval of verbal memories. Furthermore, increasing scores on verbal fluency tasks suggest that PLDP may enhance frontal lobe function and prevent or improve depressive symptoms. The mechanism is thought to involve the suppression of inflammatory cytokines produced by microglia and the increased production of BDNF. The effects observed in this study may differ from the mechanism of action of existing antidepressants, and we hope that this will lead to drug discovery as a new antidepressant. In the future, detailed studies using animal models are necessary.

The purpose of this study was to speculate on the mechanism by which PLDP improved cognitive function in participants with mild cognitive impairment, which was found to be effective in participants. As part of this effort, we discovered that PLDP may have antidepressant effects. However, this study has some limitations. First, for ethical reasons, clinical trials are open trials and the results are from a small number of participants. Second, activation of the frontal lobe by PLDP is only a possibility, as we have not directly measured the frontal lobe function. Third, it is necessary to verify whether the antidepressant effect of PLDP is true. Fourth, there was no comparison with similar drugs of the same type and efficacy. Therefore, we believe that future research is necessary to verify the results of this study in detail. 

## 5. Conclusions

This study suggests that PLDP controls subtle inflammation in the brain and enhances frontal lobe function. These findings suggest that the PLDP may prevent or alleviate depressive symptoms.

In the future, it will be necessary to examine the antidepressant and anti-anxiety effects through animal experiments. We hope that clarifying the active substances will lead to the creation of new antidepressants.

## Figures and Tables

**Figure 1 brainsci-14-00586-f001:**
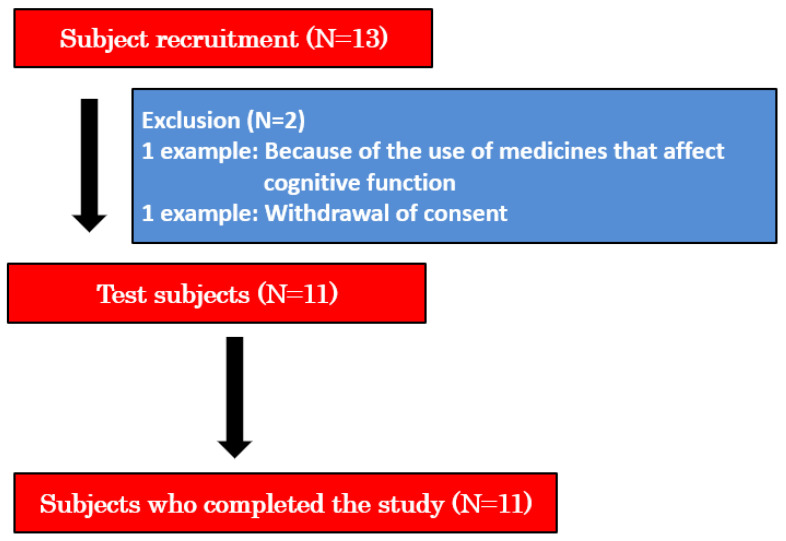
Study overview. Note. Written informed consent was obtained from all 13 participants. One of them did not participate in the study because of the exclusion criteria. Additionally, one participant withdrew consent. The test was completed by 11 participants.

**Figure 2 brainsci-14-00586-f002:**
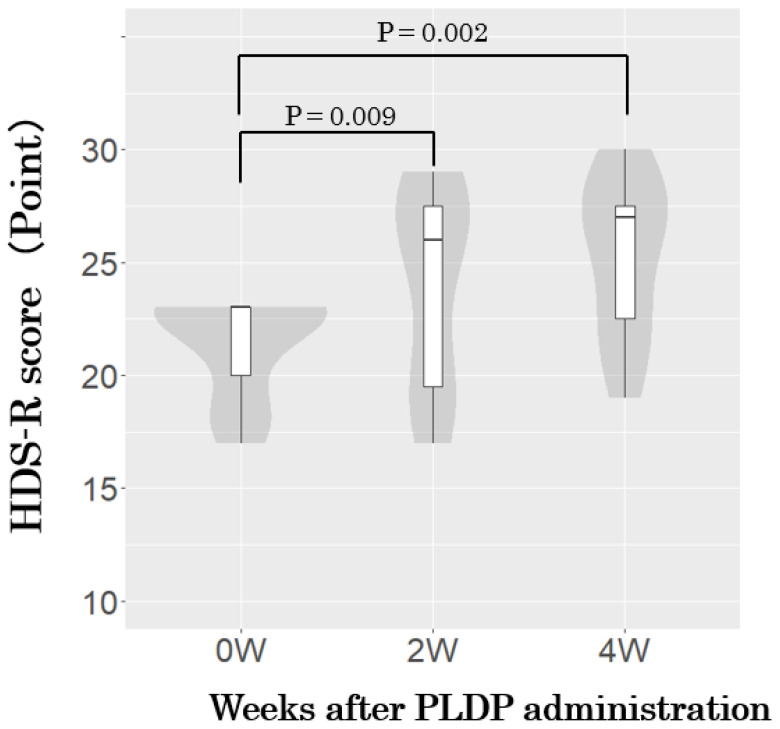
Changes in HDS-R scores after PLDP administration in participants (HDS-R: 15–23 points). Box-and-whisker and violin plots precisely represent the distribution and density of the HDS-R scores. *p* values indicate *t*-test results for scores before PLDP administration.

**Figure 3 brainsci-14-00586-f003:**
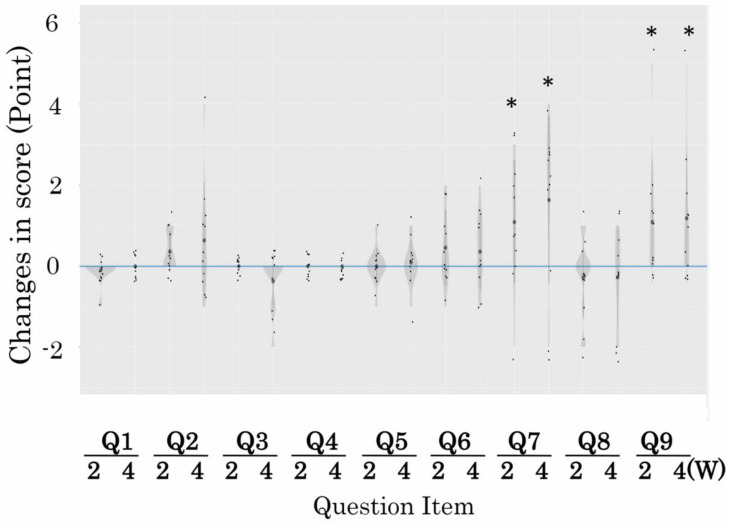
Changes in each item of HDS-R in participants (HDS-R: 15–23 points) 2 and 4 weeks after PLDP administration. Q1: Age; Q2: Orientation to dates; Q3: Orientation to place; Q4: Verbal memory; Q5: Attention; Q6: Recent memory; Q7: Delayed recall of verbal memory; Q8: Visual memory; Q9: Verbal fluency tasks. * *p* < 0.05, The *p* value is as follows. Q7 2 W is *p* = 0.0379, 4 W is *p* = 0.0251, Q9 2 W is *p* = 0.0379, 4 W is *p* = 0.0344.

**Table 1 brainsci-14-00586-t001:** Phospholipids contained in PLDP.

Moisture (normal pressure heating drying method 105 °C, 3 h)		72.90%
Crude fat (ether extraction method)		3.09%
Phospholipid (LC/MS/MS method)	Total Phosphatidylcholine (PC)	21.716 mg
	Total Phosphatidylethanolamine (PE)	2.569 mg
	Total Phosphatidylinositol (PI)	1.076 mg
	Total Phosphatidylserine (PS)	0.085 mg
	Total Phosphatidic acid (PA)	0.648 mg
	Total Lysophosphatidylcholine (LPC)	3.390 mg
	Total Sphingomyelin (SM)	0.764 mg

Moisture and crude lipids were measured at the Food Analysis Center (Tokyo, Japan). Each phospholipid was measured by Lipidome Lab Co., Ltd. (Akita, Japan).

**Table 2 brainsci-14-00586-t002:** Question items of the Revised Hasegawa Dementia Scale (HDS-R).

	Question Items
Q1	Age
Q2	Orientation to dates
Q3	Orientation to place
Q4	Verbal memory
Q5	Attention
Q6	Recent memory
Q7	Delayed recall of verbal memory
Q8	Visual memory
Q9	Word recall/verbal fluency tasks

**Table 3 brainsci-14-00586-t003:** Previous clinical evaluations of PLDP.

Study Indicator	Study Design	Participant	Result
HDS-R [1]	Study 1 Dose confirmation test (open test) (Participants: HDS-R > 20)	Low dose (N = 5) Age: 81 ± 4 High dose (N = 8) Age: 75 ± 4	No effect at 2 cap/day, significant increase at (4 cap/day)
	Study 2 Placebo-controlled double-blind study and safety confirmation study (Participants: HDS-R > 20)	Placebo (N = 25) Age: 58 ± 4 PLDP (N = 25) Age: 60 ± 4	Placebo (4 cap/day): no significant change PLDP (4 cap/day): Significant increase No change in safety
WMS-R [2]	Study 3 Placebo-controlled double-blind study	Under 40 years Placebo (N = 15) Age: 24 ± 2 PLDP (N = 15) Age: 25 ± 3 Over 40 years Placebo (N = 15) Age: 58 ± 8 PLDP (N = 13) Age: 63 ± 15	PLDP (4 cap/day): Significant score increases compared to placebo in visual memory and delayed recall (aged over 40 years)

[ ] is the cited reference number.

**Table 4 brainsci-14-00586-t004:** Participant background.

Patient No.	Age	HDS-R	Gender	Purpose of Visit
No. 1	80	23	Male	Hypertension
No. 2	75	21	Male	Prostatic hyperplasia
No. 3	53	23	Male	Hypertension
No. 4	84	22	Male	Edema
No. 5	86	23	Female	Osteoporosis
No. 6	84	19	Male	Dyslipidemia
No. 7	79	28	Female	Hypertension
No. 8	80	23	Female	Hypertension
No. 9	65	23	Female	Hypertension
No. 10	79	23	Female	Diabetes
No. 11	84	17	Female	Hypertension
Mean ± SD	77 ± 3	21.4 ± 0.7		

## Data Availability

The raw data supporting the conclusions of this article will be made available by the corresponding authors upon request.

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
