# Peer review of "Pork Liver Decomposition Product May Improve Frontal Lobe Function in Humans—Open Trial"

_brainsci, 2024, doi:10.3390/brainsci14060586_

Round 1

Reviewer 1 Report

Comments and Suggestions for Authors

The manuscript "Pork liver decomposition product may improve frontal lobe 2 function in humans - Open trial" is a search for more effective therapy in dementia diseases.

My comments:

Introduction –

What are the indications for the use of this preparation?

Material –

Were the ingredients of the PLDP preparation standardized?

Did it require storage at a specific temperature to prevent the ingredients from decomposing?

Was the preparation administered based on body weight?

Were patients informed about the composition of the preparation?

Normal aging begins above the age of 65. How the study groups were selected.

Discussion –

“Therefore, it is possible that PLDP modulates the function of microglia in the brain, enhances the function of the frontal lobe, and consequently increases the CFT scores.”- if it regulates the functions of microglia, does it participate in the production of A-beta? Is there any evidence?

If it improves memory functions, at what age should it be used, prevention or treatment?

Reviewer 2 Report

Comments and Suggestions for Authors

Title of the manuscript: Pork liver decomposition product may improve frontal lobe function in humans - Open trial

In the current open trial investigation, researchers assessed HDS-R scores at 2nd and 4th weeks of post-oral administration of PLDP by capsules, with four daily doses. Results indicated a positive trend in HDS-R score enhancement among participants. However, several limitations exist within this study. Primarily, the study was conducted early with similar HDS-R indicators and the number of targeted subjects was more limited. Majorly, the authors supported their entire study conclusions with limited pieces of evidence and followed their early experiments. Additionally, based on the limited procedures, the results' conclusion correlated with the brain’s frontal lobe functions. Furthermore, the discussion section heavily leans on earlier studies rather than the current findings.

Reviewer 3 Report

Comments and Suggestions for Authors

The article titled "Pork liver decomposition product may improve frontal lobe function in humans - Open trial" explores the potential effects of Porcine Liver Decomposition Product (PLDP) on improving frontal lobe function and cognitive decline-related depressive symptoms. This research holds significance for its potential applications and scientific value. However, there are several areas that need optimization and improvement.

1. When presenting p-values indicating statistical significance in the text and figures, it is recommended to provide specific p-values instead of just **P < 0.01 or *P < 0.05.

2. In section "2.5. Statistical analysis," the description states, "The obtained results were expressed as mean value + standard deviation," but the text actually uses mean value ± standard deviation.

3. In Figure 2, the number of data points for different weeks is not clearly or uniformly displayed, and some data points may overlap. The authors could optimize the figure to make the data points clearer.

4. It is recommended to fully discuss the limitations of this study. For example, consider discussing the sample size, study design limitations, and other relevant constraints.

Round 2

Reviewer 2 Report

Comments and Suggestions for Authors

The authors have responded to my concerns.